# Does Health-Oriented Tourism Contribute to Sustainable Mobility?

**Philipp Schlemmer [1],\*, Cornelia Blank [2] , Bartosz Bursa [3], Markus Mailer [3] and Martin Schnitzer [1]**

[1] Department of Sport Science, University of Innsbruck, Innsbruck 6020, Austria; Martin.Schnitzer@uibk.ac.at
[2] Institute for Sports Medicine, Alpine Medicine and Health Tourism, UMIT, Hall 6060, Austria; cornelia.blank@umit.at
[3] Department of Infrastructure, Unit of Intelligent Transport Systems, University of Innsbruck, Innsbruck 6020, Austria; Bartosz.Bursa@uibk.ac.at (B.B.); Markus.Mailer@uibk.ac.at (M.M.)
\* Correspondence: Philipp.Schlemmer@uibk.ac.at.com; Tel.: +43-512-507-45801

**Abstract:** Previous research has concentrated on traffic and health-related effects in relation to arrival patterns (e.g., stress in connection with means of transport, travel distance, and time). However, tourist mobility behavior during a vacation and potential health-related associations as well as its correlation with physical activity habits and mobility patterns at home seems to have barely been examined. To address this research gap, the study at hand applied a mixed-method approach in three tourism destinations in Tyrol, Austria. The study examined tourists' mobility as well as their physical activity at home and at the holiday destination. Results show that the preferred transport mode (arrival and on-site) is the private car, due to its flexibility and comfort. Hotel front desks, as the main information source, determine tourists' behavioral aspects during a vacation. General mobility routines show differences between everyday life and holiday situations, and physical activity is important for the overall satisfaction of tourists, which proves to be more intense and frequent during a vacation than in everyday life. Seven percent of the tourists participating in the research stated that they had changed their daily mobility behavior after their return, with most of them walking or cycling more often. This study contributes to research in the field of sustainable tourist mobility and physical activity behavior and highlights the necessity for further specific analyses.

**Keywords:** health-oriented tourism; active mobility; physical activity; sustainability

## 1. Introduction

Walking and cycling as modes of transport, also referred to as active mobility (AM), are of considerable importance for sustainable mobility. On the one hand, increasing AM reduces the consumption of space for transport infrastructure, energy consumption, air pollution, and noise [1–4], meeting ecological goals of Sustainable Urban Mobility Plans [5] and strategic EU policy documents [6–8]. On the other hand, AM provides mobility at comparably low cost for travelers and the public, thus meeting economic goals as well. Furthermore, AM can be used by most people without restriction, it is affordable and accessible, and does not represent an accident risk to others. In addition, AM is healthy in that it provides regular physical activity [1]. Hence, AM also meets social aspects of sustainability, thus playing an essential role in visions and strategies for sustainable mobility (e.g., [9]).

However, mobility surveys show that the share of AM in daily travel has been decreasing in recent decades (e.g., [10,11]) for various reasons. Although the promotion of cycling in some cities has succeeded in significantly increasing the share of cycling [12], this has not been achieved in rural areas and the proportion of walking has decreased significantly. Increasing people's awareness of health and

fitness can provide an opportunity to increase AM, yet AM is not only well suited to providing regular physical activity. In contrast to sports or exercise, AM requires less time and less motivation [1]. It can easily be integrated into daily routines and it is economically affordable. It, therefore, also has the potential to reach sections of the population who may be less willing to engage in sports and exercise, or who cannot afford to do so due to monetary restrictions or lack of time [13]. Thus, some authors assume that, for people with low levels of physical activity, such as sedentary, obese and elderly people, it is easier to use AM as a moderate form of regular physical activity than to take up sports or other types of vigorous physical activity [13,14].

Detailed studies on the inter-relationship between AM and physical activity are seen as unsuitable [13], since the majority give an incomplete picture of AM due to methodological limitations that either focus on mobility or physical activity. The European research project PASTA (Physical Activity through Sustainable Transport Approaches) investigated correlates and inter-relationships between AM and physical activity and evaluated the effectiveness of selected interventions and measures to promote AM so as to increase both AM and physical activity [1]. As a result, a handbook of good practice for the promotion of walking and cycling was published based on case studies in several partner cities. The description of the case studies and recommendations for local authorities presented in the handbook include strategic policies, physical environment and infrastructure, social environment, as well as regulation and legislation in the fields of transport and health (www.pastaproject.eu). However, even though PASTA addresses the inter-relation between AM and physical activity and renders visible the links between transportation and health, its focus is only on daily mobility in big cities (case study cities are Antwerp, Barcelona, London, Örebro, Rome, Vienna, Zurich). It does not provide data about the interrelation between AM and physical activity levels on a personal basis with a view to showing the potential of changing mobility patterns.

One of the limitations of modifying daily mobility behavior is that it is very routine. Therefore, situations that break with daily routines or lie outside these routines are seen as an opportunity for the initiation of behavioral changes. Vacations can be considered as representative of such situations. So, mobility at tourist destinations especially at destinations which promote health-oriented holidays linked with physical activity could have the potential to promote AM. However, to the best of our knowledge, there is no detailed research into the question of whether vacations, including an active program, can be an effective means of triggering a change in daily AM at home. Hence, this paper aims at analyzing health-related mobility and transport patterns of tourists on a personal level at home and at their holiday destination. In this context, tourists were surveyed about modes of transportation, the reasons for selecting the means of transport, as well as possible differences in their daily and vacation-specific mobility patterns. We, therefore, aim to address the above-mentioned gap and to identify possible policy-related implications.

A comprehensive review of the mobility and tourism literature, determining the physical activity factors during a vacation that influence sustainable mobility, is provided in the next section. The research questions used in this study are described in the subsequent section, followed by the method section. The findings of the study are presented thereafter. Finally, the discussions, limitations, conclusions, managerial implications for destination managers and policy-makers as well as an outlook for future research are presented.

## 2. Theoretical Background

Generally, tourism consists of all activities, such as recreation, business, family visits or events that people undertake during their trips away from their habitual environment [15]. Tourism consists of manifold characteristics, whereby transport and mobility are more often linked to the journey itself than the on-site movement of tourists at certain destinations. In the recent past, some research has considered tourist mobility as an essential public issue in dedicated tourism regions as well as the contribution of tourism to climate change, with transportation to/from the destination being recognized as a primary contributor [16–21]. Agility, quality, and rapidity of tourist mobility also influence the

competitiveness of destinations, through enhanced tourist satisfaction [22–25]. Scuttari et al. [26] address the tourism–traffic paradox in ecologically sensitive but tourism-intense areas, where transport policy-makers understand the need to minimize transport-related impacts, yet are practically unable to change the status quo fearing negative effects of traffic management on the tourism industry. However, previous concerns about the dominance of economic or cultural aspects of tourism have recently altered in favor of thoughts about sustainable tourism, balancing economic, social as well as ecological intentions [27] including concepts of sustainable and active mobility [28]. Without going to deep into the topic of sustainable tourism, which goes beyond the scope of this paper, it can be said that local mobility at a destination has a decisive influence on the mode chosen for the journey to the destination and thus on the sustainability of tourism. Similar to the perspective of Hunter ([29], p. 860ff), tourism can be seen as a catalyst for sustainable development, and as one of the most accessible drivers of change and improvement not only at tourist destinations. Given these considerations, one could think of the health-related mobility characteristics of tourists at a destination as the sustainable specification of mobility and transport.

### 2.1. Tourist Mobility Behavior

As already mentioned, like daily activities when at home, tourist activities on vacation strongly depend on mobility and accessibility [30,31]. Tourism development necessitates travel and mobility of tourists, whereby travel and mobility are recognized separately, where travel constitutes the journey to a destination and mobility pertains to transportation at a tourism destination [32]. The trend toward more holiday trips to increasingly remote destinations affects the demand for travel [33], simultaneously triggering a double-edged evolution toward higher economic and social benefits, on the one hand, and an ecological and environmental liability for the destinations, on the other. The transportation of tourists alone accounts for 86% of the entire environmental impact of tourism [34]. Nearly 75% of holiday trips to Austria are made by private car [35]. They have a substantial influence on the levels of traffic congestion on alpine roads, generating pollution in these environmentally fragile areas and even disrupting the daily activities and travel patterns of local inhabitants, who experience severe difficulties in traveling due to a lack of alternative roads in the densely built-up and narrow mountain valleys [26].

The scientific literature within this area merely concentrates on long-distance travel to and from a destination, and analyzes the decision-making processes [36], trip destination, and transport mode choices [37–39] or environmental impacts [40,41]. Some specific aspects of intra-destination mobility are also covered, be it the use of public transportation [42,43], mobility in rural areas [26,44], movement patterns [45,46] or access policy measures [47]. However, we have not identified any analyses of tourists' mobility behavior on-site that focus on the relationship between physical activity habits and mobility patterns in daily life, which inspired us to conduct this study.

Within the context of sustainability, researchers often differentiate between tourist types clustered according to leisure activity and mobility behavior [48,49]. The typology ranges from "fun and actions seekers" to "sophisticated cultural travelers", distinguished by the degree and direction of sustainability in their context or a more pragmatic classification by travel distance and total number of holiday trips [40]. Böhler et al. conclude that "strategies aiming at the reduction of the individual's negative environmental impact have to consider different personal preconditions for travelling as well as the different extent to which people travel" ([40], p.666). A deeper understanding of the leisure aspect of travel can be substantiated by specific leisure mobility styles [50,51].

### 2.2. Tourist Health-Related Activity

With the exception of typical sports activities (e.g., skiing, hiking) practiced by visitors to Austria for many decades, more specialized opportunities have emerged in recent years covering trail running, climbing, backcountry skiing, alpinism, MTB, road cycling, etc. [52,53]. Moreover, not only is the amount and variety of sports-focused resorts and accommodation higher, but the number of wellness,

health, and recovery options aimed at less sporty visitors is on the rise, too [54]. There is a broad selection of literature on the impacts of this trend on destination management [55], hoteliers, and the hospitality industry. Also, the effects of a vacation on health and wellbeing have been investigated in considerable depth [56,57]. It has been confirmed that there is a noticeable improvement in health and fitness levels after a vacation, even though these effects do not persist for long after returning home and tend to decline within days or weeks [58].

Surprisingly, what has not been investigated thus far is whether the physical activity and health-awareness of tourists has an effect on their travel behavior, which is inseparably connected to the performed activities. This is all the more important, as most tourist activities require travel to access the activity start locations, specific infrastructures or facilities. There is also no evidence of health considerations in mobility and transportation policies that would sustainably influence the environment and people's health [59]. This is supported by Higham et al. [60], who encourages the consideration of health aspects in sustainable tourism, mobility, and transport.

## 3. Research Questions

Considering the interdependencies between tourist health-related activity and mobility behavior, we have put forward the following four research questions having hypothesized differences in each RQ:

- RQ1: How is travel behavior on vacation different from daily life? What is the role of sports activity?

We were particularly trying to understand whether physical activity may be a motivation for the use of active transportation modes (cycling, walking) on vacation or in daily life. Furthermore, the study should provide answers as to whether or not health/sports-oriented tourists are more likely to lean toward active transportation. Finally, RQ1 should shed light on whether the use of active transportation replaces sports and fitness activities.

- RQ2: Do the mobility and activity patterns developed during a vacation persist after returning home and trigger more sustainable daily mobility?
- RQ3: Why do guests decide to use a particular transportation mode for arrival and at the destination?

We particularly aimed at understanding whether tourists' choices in their means of transport are driven by health or sustainability considerations? Finally, RQ3 aims at understanding what motivates people to use more sustainable modes of transport at the destination?

The abovementioned research questions (RQ1–RQ3) were posed using a mixed-method approach and the data obtained were thoroughly examined with mostly descriptive, statistical methods.

## 4. Data and Methods

### 4.1. Contextual Background

The study was carried out in the federal state of Tyrol in Austria. Tyrol is an alpine region characterized by its dependency on tourism, focusing particularly on sports tourist activities, both winter (alpine skiing, back-country skiing) and summer (hiking, cycling, climbing) disciplines. Tyrol is currently positioning itself abroad as a region for sports tourism organizing sports events (e.g., UCI Road Cycling Championships) and is constantly developing its offer of physically engaging activities (e.g., e-biking) [61]. Additionally, the fact that Innsbruck, the capital city of Tyrol, displays a high share of cycling in public transport secondarily strengthens the study region of Tyrol. Since our research focuses on people choosing health-oriented holidays with higher levels of physical activity and active mobility, Tyrolean destinations are ideal locations when looking for answers to our research questions. To gather a more detailed overview of the mobility-related situation in the federal state of Tyrol, three characteristic tourist destinations (Innsbruck and its holiday villages, Pitztal and Hohe Salve) are considered in our study, reflecting the current tourism situation across the entire region of Tyrol.

### 4.2. Study Design and Procedures

The current study utilized a mixed-method approach including focus groups (step 1) and a quantitative questionnaire (step 2), including tourists in three Tyrolean holiday destinations (Hohe Salve, Innsbruck, and Pitztal). To ensure content validity of the questionnaire, results of the focus groups informed the design of the questionnaire, as no validated instrument was available to analyze the research questions. The study at hand was carried out during the period from May 2016 to February 2017 and was supported by each participating holiday destination. Participants in the focus groups were based on a convenience sample and selected by the respective participating hotels. The only inclusion criterion was a previous sojourn by the participants in the respective region. The hotels themselves were recommended by the respective destination management organization.

#### 4.2.1. Qualitative Analysis

Prior to conducting a quantitative analysis, six focus group interviews were carried out with holiday guests from all three study regions. The aim of these workshops was to examine the mobility behavior of the guests at the resort and at home, as well as movement and health aspects. This qualitative approach served to better understand the needs of guests and their behavior in order to provide a basis for the subsequent quantitative survey. The workshops took place in five hotels (2× Hohe Salve, 2× Innsbruck, 2× Pitztal), which were recommended by the respective destination management organization. The workshop participants were approached and selected by the specific management of the hotel, fulfilling the requirement that they had spent at least one previous vacation at one of the specified holiday destinations. The qualitative focus groups discussed (health-related) mobility and transportation patterns in daily life and during a vacation as well as their possible mutual impacts. The workshop took place in a moderated form and was structured by means of two flipcharts on the topic of "mobility", as well as "sporting activities". Here, the participants were asked to mark the categories applicable to them with a glue dot. The resulting mood of the participants was then discussed in the group in order to gain deeper insights into the topics of (health-related) mobility, transport, and satisfaction. The focus groups were conducted in English, German or Italian language—depending on the nationality of the participants. All focus groups were carried out according to the ethical guidelines and criteria stated by Patton [62]. Afterwards, the focus groups were transcribed verbatim in their original language and the raw material was read and encoded, following the qualitative content analysis of Mayring and Fenzl [63].

#### 4.2.2. Quantitative Analysis

The online survey was conducted between 1 December 2016 and 28 February 2017. The questionnaire consisted of three general topics, concerning socio-demographic data, activity-related interests, as well as mobility patterns in everyday life and in holiday settings. The tourists were approached through the newsletter of each tourism region, which included a link to the specific questionnaire of each holiday destination. In detail, all tourists were asked to make their specifications regarding their last holiday (regardless of whether it was a winter or summer vacation) in one of these three Tyrolean holiday destinations. The study at hand was conducted according to the "ethical guidelines for surveys" approved by the Institutional Review Board (IRB) of the Department of Sport Science as well as the Board for Ethical Issues of the University of Innsbruck. Socio-demographic data assessed included information about gender, age, level of education, income, occupation, origin, and level of physical activity.

Mobility behavior included questions about everyday life mobility, on-site mobility as well as decisive motivational patterns. The structure of daily mobility was captured in the questionnaire by asking about frequency of use of a particular transport mode. Seven transport mode categories were distinguished: walking, cycling, e-biking, public transport (bus, train, etc.), private car/motorbike as driver, private car/motorbike as passenger and car-sharing. Furthermore, five frequency categories

were applied: 4–7 days per week, 1–3 days per week, 1–3 days per month, more rarely, never. Based on these use frequencies, two level categories: actively mobile (more than 1 day/week) and passively mobile (less than 3 days/month) were created for each transport mode and the respondents were grouped into one of them. Mobility on vacation was measured by asking the respondents which transport mode out of the following 11 they used as the main mode at the destination: private car, rented car, campervan, motorbike, coach, train, bus, taxi, bicycle, e-bicycle, walking.

The level of physical activity was assessed by using the Godin Leisure Time Exercise Questionnaire, categorizing respondents into three groups. The categorization was conducted via the proposed Formula (1), resulting in the Godin Scale Score:

$$\text{Weekly leisure activity score} = (9 \times \text{Strenuous}) + (5 \times \text{Moderate}) + (3 \times \text{Light}), \qquad (1)$$

The values for strenuous, moderate and light activities were addressed in the question: how many times on average do you engage in the following kinds of exercise (strenuous exercise—heart beats rapidly; moderate exercise—not exhausting; light exercise—minimal effort) for more than 15 min during your free time in a typical 7-day period (a week)? The given information about strenuous, moderate, and light activities was multiplied by nine, five, and three respectively and further divided into the group of Insufficiently Active/Sedentary people (<14 units), Moderately Active people (14–23 units), and Active people (>24 units).

In total, 588 persons participated throughout the period of the survey, with 127 tourists from the holiday destination Hohe Salve (36.6% female; 63.7% male), 402 tourists from the holiday destination Innsbruck and its holiday villages (57.4% female; 42.6% male) and 59 tourists from the holiday destination Pitztal (43.1% female; 56.4% male). The minimum and maximum ages were 18 years and 81 years, resulting in an average age of about 46.4 (14.1) years. The average number of overnight stays was 4.6 (4.2) days.

*4.3. Statistical Analysis*

All statistical analyses were carried out using SPSS v. 24.0 (IBM Statistics, IL, United States). Relative to the research questions, Chi2-Tests, linear mixed-model analyses in the case of homogeneity of variances with Bonferroni-corrected post-hoc tests, as well as Welsch tests in the case of variance inhomogeneity with Tamhane-corrected post-hoc tests, were performed. Core outcomes were the main sociodemographic data, the main effect of physical activity over time, the interaction effects between activity level and holiday destination as well as effects according to the specific mobility patterns. In the case of non-significant interaction effects but significant main effects on well-being, Sidak-corrected or Tamhane-corrected post-hoc tests were used to provide more details on the effects. In the case of a significant interaction effect (activity level), post-hoc ANOVA was performed separately for the respective groups. The level of significance was set at $p < 0.05$ with the additional information about the effect sizes (Eta-squared). Unless otherwise stated, data are presented as mean values (MV) with standard deviation (SD) or relative frequencies (%).

## 5. Results

*5.1. General Findings Relating to the Sociodemographic Profile and Activity Levels*

At the beginning, it was striking that the study showed very large intra-regional differences in the number of questionnaires answered. As already mentioned, the questionnaire was completed by 127 persons in the holiday region Hohe Salve, 402 persons in Innsbruck and its holiday villages, and 59 persons in the holiday region Pitztal. The number of completed questionnaires differed significantly between the holiday regions. The distribution of gender in the questionnaire also showed a very significant difference (Chi (2) = 18.67, $p < 0.001$).

The average age of the guests in the holiday regions also differed significantly (F (2141.18) = 21.98, $p < 0.001$, Hohe Salve 54.17 ± 11.71 years, Innsbruck 46.26 ± 14.08 years, Pitztal 53 ± 13.82 years),

with the Tamhane post-hoc test showing highly significant differences between the holiday region Hohe Salve and Innsbruck ($p < 0.01$), as well as Innsbruck and the holiday region Pitztal ($p < 0.01$). The holiday frequency in the respective holiday region also differed significantly (F (2240.23) = 11.99, $p < 0.01$). In the Hohe Salve holiday region, the guests surveyed had already been 6.75 ± 11.45 times, compared to 5.02 ± 12.08 times in the Innsbruck holiday region and its holiday villages, and 2.16 ± 3.61 times in the Pitztal holiday region, differing significantly according to post-hoc analysis (Hohe Salve/Pitztal, $p <0.01$; Innsbruck/Pitztal; $p < 0.01$).

With regard to length of stay, significant differences can also be observed (F (2565) = 3.95, $p = 0.02$), with guests spending an average of 6.64 ± 7.91 days in the Hohe Salve holiday region, 4.94 ± 5.37 days in Innsbruck, and 5.05 ± 3.81 days in the Pitztal, with the Bonferroni post-hoc test only showing the difference ($p = 0.017$) between the Hohe Salve and Innsbruck regions. The tourists' satisfaction levels with the specific hotel offering did not differ significantly between the regions. By contrast, the regions differed in the usage behavior of guest cards, with 60% in the Hohe Salve holiday region and 71% in Innsbruck and its holiday villages using a guest card (Chi2 = 4.94, $p = 0.026$). The summer map was also used significantly differently within the regions (Hohe Salve 83%, Innsbruck 65%, Pitztal 66%, Chi2 = 13.3, $p < 0.01$).

According to Godin [64], the physical activity of the tourists was significantly higher on vacation (61.62 ± 113.71) than at home (48.7 ± 132.16; T (350) = −3.63, $p < 0001$); however, there was no significant interaction effect ($p > 0.05$), confirming the similarity of destinations (Table 1). This significant increase in physical activity can be observed in all holiday regions ($\eta2 = 0.036$, $p < 0.01$). The vacation duration shows no relation to the extent of physical activity at home or on vacation ($p > 0.05$) and also the inner-regional view did not lead to a significant result ($p > 0.05$). In general, the extent of physical activity while on holiday did not show any gender, age, education or income-related differences, and vacation planning had no effect ($p > 0.05$). Satisfaction with the activities on offer at the tourist accommodation of the three holiday regions does not affect the extent of physical activity.

**Table 1.** Overview of mean values and standard deviations of Godin scores according to holiday region.

| Score | Holiday Region | Mean Value | Standard Deviation |
|---|---|---|---|
| Godin score in everyday-life | Hohe Salve (n = 67) | 39.7 | 28.7 |
| | Innsbruck (n = 244) | 51.3 | 156.9 |
| | Pitztal (n = 40) | 47.9 | 41.7 |
| | Total (n = 351) | 48.7 | 132.2 |
| Godin score in holiday setting | Hohe Salve (n = 67) | 65.8 | 73.9 |
| | Innsbruck (n = 244) | 60.2 | 129.7 |
| | Pitztal (n = 40) | 63.4 | 44.5 |
| | Total (n = 351) | 61.6 | 113.7 |

*5.2. Specific Findings Relating to the Research Questions*

Regarding RQ1, the results reveal that the predominant mode of transport used for on-site mobility while on vacation is the private car (46.2%), followed by bus (20.0%), walking (18.6%), and e-bike (6.6%). A similar picture can be painted in the use of various transport modes in daily life (Figure 1). Driving a private car and walking are most common, followed by public transport and cycling.

Our initial hypothesis that there might be a correlation between daily and vacation mobility patterns was not reflected in the data. Using Chi-square and Fisher tests between vacation/everyday-life and active/passive utilization, we found that there was no association between how often people used particular transport modes in daily life and whether they then decided to use motorized or non-motorized modes of transport while on vacation. No differences between groups were found to be significant.

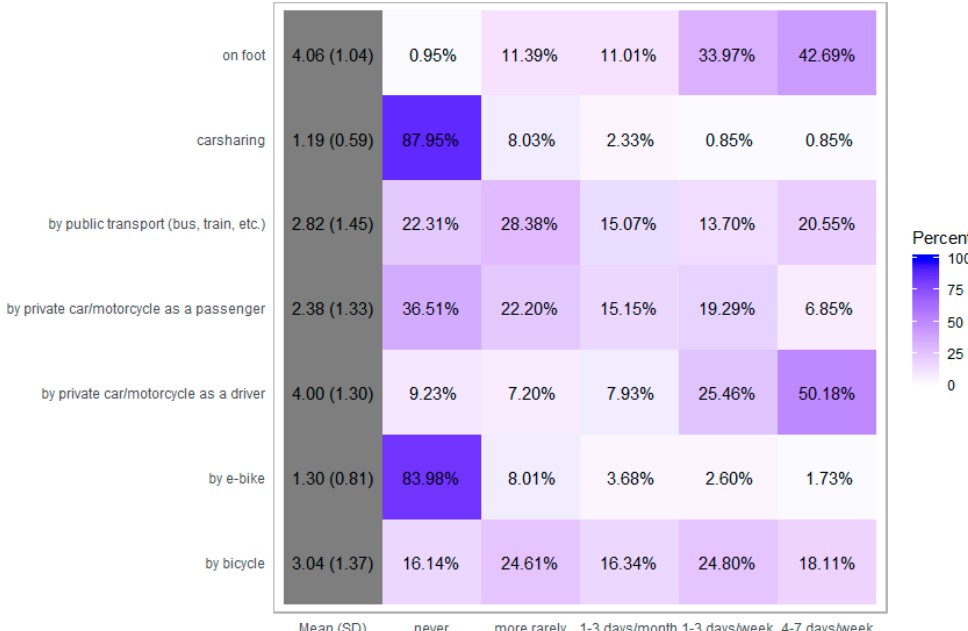

**Figure 1.** Heat map of choice and frequency of means of transport in daily life.

A further aspect within this research question was to determine whether there were differences between the sport activity groups in their choice of non-motorized (active/sustainable) and motorized transportation at the destination. The Chi-square test returned a value of 4.484 at the $p$-value = 0.106, indicating no different distribution with regard to the choice of mobility.

A similar analysis was conducted to search for relationships between daily mobility and sports activity levels. The Pearson Chi-square test was used to investigate whether there were significant differences between the three activity level groups for each of the seven transportation modes used in daily life (Table 2). The significant differences were found in three groups: walking, cycling, and using a private car/motorbike as a driver.

**Table 2.** Differences due to the activity level and transportation mode in daily life; significant differences are highlighted in bold type.

| Transport Mode | Chi-sq Value | *p*-Value |
|---|---|---|
| **Walking** | **27.51** | **<0.001** |
| **Cycling** | **26.96** | **<0.001** |
| E-biking | 1.94 | 0.38 |
| Public Transport (bus, train, etc.) | 2.13 | 0.34 |
| **Private car/motorbike as driver** | **6.93** | **0.03** |
| Private car/motorbike as passenger | 5.52 | 0.06 |
| Car-sharing | 1.58 | 0.45 |

Looking at the groups split within walking, we can observe a high share of sports-active respondents also being active walkers. However, the results only indicate that people in the sedentary group were significantly less often (than expected) in the walking-active group and significantly more often in the walking-passive group (at $p < 0.05$). This means that the sedentary respondents walk less, and this relationship is significant.

There is a substantial difference in the use of bicycles for transportation among the sedentary respondents, while the active and moderately sports-active fall into the active and passive user groups quite evenly. The sedentary respondents were significantly under-represented (at $p < 0.001$) in the cycling-active group and significantly over-represented (at $p < 0.01$) in the cycling-passive group, which implies that the insufficient level of sports activity in daily life goes hand in hand with only very

rarely using a bicycle for transportation (less than 3 days in a month). No other differences between groups were found to be significant.

As far as driving a private car is concerned, one can still observe that the majority of drivers are physically active in daily life. However, it can be reported that the sedentary respondents fell significantly more often (at $p < 0.05$) into the passive-drivers group, which means that being insufficiently sports-active in daily life can result not only in walking and cycling rarely but also in driving rarely. No other differences between groups were found to be significant.

Regarding RQ2, we analyzed whether mobility and activity patterns developed during a vacation persisted after returning home and triggered more sustainable daily mobility. We found that 93% of respondents in all three physical activity groups did not change their daily mobility habits after coming back home from a vacation. However, among the 7% who changed their habits, 73% declared that they had started using more AM, predominantly walking more, and nearly one-third started traveling more by public transport/car-sharing.

As far as RQ3 is concerned, we found that the majority (74%) of respondents specified that their private car had been their main transport mode for the trip to the vacation resort. The main reasons for arriving by car are, above all, on-site mobility, flexibility, luggage transportation, comfort, and travel time. Besides the exact reasons for choosing a particular mode of transport for arrival or for traveling at the destinations, we also examined their attitudes towards sustainability in transport. Figure 2 depicts the demand for necessary conditions that would have to be fulfilled to encourage visitors to use environmentally friendly modes of transport on vacation. There is a broad acceptance for making a positive contribution to climate and nature preservation. However, over 80% of respondents are not willing to compromise on flexibility and comfort while traveling. They also want to be rewarded (with discounts, vouchers, etc.) for their environmentally friendly behavior. One can observe a visible resistance to undertaking a leading role in the process. On the other hand, almost a third of respondents do not mind being the only ones using the environment-friendly transportation modes.

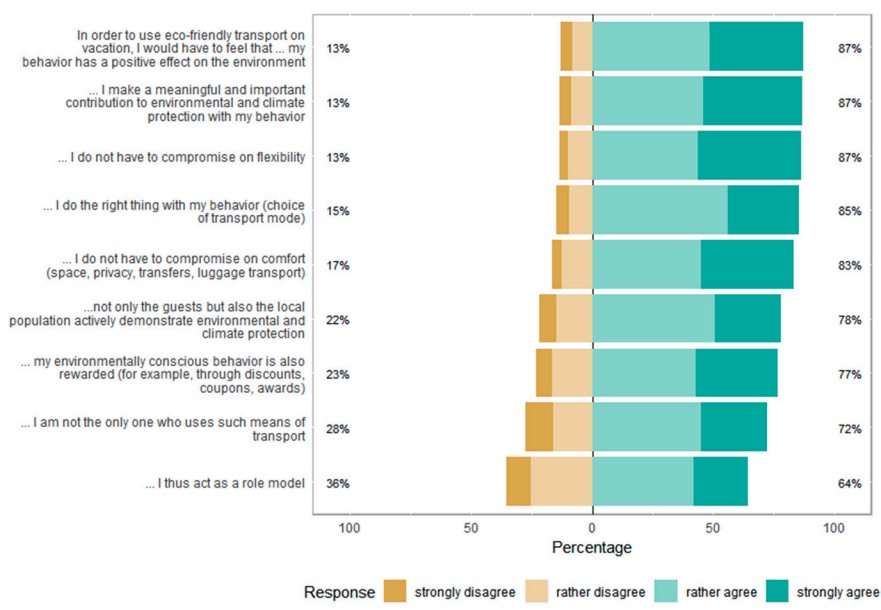

**Figure 2.** Preferences of sustainable mobility behavior.

## 6. Discussion

### 6.1. Tourist Transportation and Mobility

On-site mobility on vacation proves no direct relationship with the level of physical activity. As far as daily mobility is concerned; however, relationships do emerge. The respondents engaging in more physical activity in daily life also tend to walk and cycle more for everyday transportation. Similarly,

the sedentary respondents walk and cycle less, and even drive significantly less. This allows us to conclude that while the level of physical activity at home might influence daily mobility patterns, the mobility behavior on vacation is driven by other factors than the level of physical activity on-site. Consequently, while, for example, promoting physical activity or building sports infrastructure in the regions from which the visitors to the Alps usually originate (non-tourist urban/metropolitan areas) might result in higher shares of sustainable, non-motorized transportation modes in the modal split, this would not be the case at tourist destinations. Surprisingly for us, the mobility patterns at home seem to be unrelated to the ones adopted during a vacation. Furthermore, the active cyclists/walkers at home do not cycle/walk significantly more on vacation, nor are the daily drivers driving more at their vacation resorts.

These travel decisions are made independently in different conditions and considering different factors. A limitation of the questionnaire was that the questions about daily mobility pertain to an average week in the respondent's life, which means that we cannot distinguish whether the respondent was reporting about the week before or after their vacation. We could not, therefore, investigate the direct relationship between physical activity on vacation and daily mobility after returning home. It is only possible to assess this indirectly by asking if their travel behavior changed after the vacation. The few respondents in the sample that did change their daily mobility behavior claimed to be walking or cycling more. This, however, was found to be unrelated to physical activity levels, which could for instance mean that the physically active vacationers did not start using non-motorized transportation after their vacation significantly more than before.

Apparently, there are other, more influential factors shaping mobility decisions during a vacation, such as flexibility, luggage transportation and on-site mobility. This is supported by the answers to the question regarding attitudes towards sustainability. The respondents do not want to give up comfort and flexibility when traveling on vacation, even though their overall awareness level about sustainability, environment and climate is high. In the context of health-oriented behavior, particular attention should be paid to the promotion of the use of physically active modes of transport (walking, cycling or e-biking). The urge for active travel such as cycling, walking or taking non-motorized means of transport is rising, which is also mentioned in literature as significant cardiovascular effort or beneficial metabolic costs [65]. The analysis of active travel also implies economic value for recreational and non-recreational interests, which have already drawn political interest, leading to integration in the operations of public transport organizations, the promotion of tourism, leisure, and sport as well as the integration of active travel by the public [65] and in integrated planning strategies for sustainable tourism [48]. A recent study from Norway shows that e-bikes are of little interest to people who already utilize bicycles for personal transport or even for exercise. However, e-bikes can serve other purposes [66]. Active modes of transport are often underestimated in terms of their public value, whereby promoting these active modes of travel can reduce expenditure on transportation for individuals and communities [67]. This means that, the way one travels on vacation and the way one travels at home seem to be independent decisions.

Additionally, the acknowledgement of active travel is rising in society due to its impact on human health [68], substantiating the importance of active travel. Furthermore, active travel contributes to a sense of vitality, social cohesion, and sustainable tourism, with an associated aim of reducing dependence on car travel [69,70]. This is partly motivated by a desire among people who want to travel in a sustainable manner and want to be less dependent on motorized means of transport. These people are also aware of the cost-efficient and sustainable characteristics of active travel. Important factors in promoting public cycling are not only infrastructural improvements but also information and marketing. However, since the general traffic situation in the municipalities also plays a significant role, the supply and motivation of the local population and those employed in tourism are also essential [67]. In particular, there is room for improvement in the availability of information on accessibility to public transport and the sometimes very diverse modes of transport in the respective regions.

### 6.2. Physical Activity

The regions were specifically selected for physical activity on vacation, because tourists are aware in advance of the sporting activities on offer, represented by the frequency of physical activities while on vacation ($61.62 \pm 113.71$). This would also explain the generally high level of satisfaction in all regions.

These results are backed by findings from previous research done by De Bloom et al. [71], who tried to answer the question of possible activity effects during a vacation on satisfaction and wellbeing. Time for physical, social, and passive activities was considered in their analysis and they revealed that engagement in physical, social, and passive activities contributes to enhancements in satisfaction, wellbeing and health during and after a vacation. Furthermore, they show that time spent on physical activities has a strong effect on these parameters, which means the more time spent on physical activity, the better the effects on satisfaction, health and wellbeing during a vacation. However, this cannot be taken for granted as the results of De Bloom et al. [72] show. This study states that activities and experiences in vacation situations are only weakly associated with improvements in health, satisfaction, and wellbeing. Only, passive activities, savoring, relaxation and sleep are determining factors for health and wellbeing improvements. There is general consensus that (physical) activities are seen as important influencing and predicting variables for satisfaction. Contemporary research, though, highlights impacts of physical activities during a vacation on satisfaction, wellbeing and other health parameters [57]. Interestingly, there are no regional differences between destinations. One may, therefore, assume that city tourists are less sports-affine than the guests in the other regions. Here, perhaps, the image of Innsbruck as a sports city could also have attracted more sporty tourists.

### 6.3. Limitations

Finally, several limitations of this study should be mentioned. First of all, so far only the federal state of Tyrol has been a focus of research and was only represented by three different tourism destinations. Secondly, we found only a few voluntary participants for the focus groups, which results in very limited outcomes for the qualitative part of the study, and furthermore, the convenience sample may constitute a limiting factor. Moreover, the survey was conducted via the newsletter services of each holiday destination, which also only reaches a certain community and may lead to a bias. A further limiting factor is limited regional examination of respondents and a possible restriction of the informative value of the current study, which stands alongside the conscious decision to use three different holiday destinations in Tyrol for transparency and comparability. This limitation clearly provides the basis for further discussion.

## 7. Conclusions, Implications and Outlook

The outcomes of this research revealed several implications that are worthy of consideration by tourism destinations as well as policy-makers. As sustainable transportation and mobility are the crux of prospective tourism planning, more effective knowledge about behavioral tourist patterns, arrival patterns etc. of tourists is essential. Active mobility patterns are seen as a simple way to include beneficial physical activity into people's daily lives, whereby active mobility or transportation is understood as any kind of transport movement requiring human physical power [72]. As a function of accessibility, active transportation modalities are crucial for the selection of the means of transport as well as the layout of the community [64]. In this context, Handy and Clifton [73] reflect upon accessibility as the simplicity to reach desired activities without the utilization of motorized means of transport. Furthermore, the infrastructural circumstances can facilitate or constrain physical activity. Saelens, Sallis, and Frank [74] claim that the relationship between the built environment and physically active means of transport or driving have to be differentiated. Previous research further agrees that the design of community also supports the role and choice of the means of transport used [75]. Density and intensity of development, the functionality of destinations, the aesthetic qualities of certain destinations and the connectivity of the street network are characteristics of activity-friendly

destinations or communities [74], otherwise known as walkable communities [76]. Additionally, research shows that residents of pedestrian- or cycling-oriented destinations engage in more physically active transportation than other communities [77]. This suggests that the residents' exemplification of an active lifestyle as well as the clear preference for active means of transport for tourists are taken into consideration.

Building upon the background of current touristic developments, which indicate a rise in the recreational interests of tourists, offers including physical activities and sustainable active mobility could potentially add a unique facet to a holiday destination, both for tourists and residents.

Unfortunately, the findings suggest no direct relationship between the level of physical activity during a vacation and mobility behavior. There is, however, potential for future change as 7% of respondents declared they altered their behavior toward AM after the vacation, with the majority starting to walk and cycle more, and many using public transport more frequently. Taking into account how difficult it is to initiate behavioral changes in transportation by traffic measures due to the routine nature of transport, this group of 7%, though apparently small at first glance, is in fact quite remarkable and could potentially become a starting point for more far-reaching changes. This shows that the promotion of AM can build on the potential of health-oriented motivations. Campaigns should therefore intensify their focus on health and life-style messages alongside environmental reasons. Moreover, the promotion of more active means of transport during leisure and recreation time may also provide a more active and sustainable mobility pattern in daily life. We suggest further analysis in future research (e.g., to what extent can the frequency with which AM is used increase after a vacation).

Nevertheless, there is still no definitive understanding of active mobility patterns among tourists in alpine destinations, which therefore requires further investigation. It is hoped that the present findings can be taken into account for further and future research in the field of sustainable tourism and transportation.

**Author Contributions:** The following authors contributed to conceptualization, P.S., B.B., C.B., M.M. and M.S.; methodology, P.S., B.B., C.B., M.M. and M.S.; formal analysis, P.S., B.B., C.B., M.M. and M.S.; investigation, P.S., B.B., C.B., M.M. and M.S.; writing—original draft preparation, P.S., B.B., C.B., M.M. and M.S.; writing—review and editing, P.S., B.B., C.B., M.M. and M.S.; visualization, B.B.; supervision, P.S., B.B., C.B., M.M. and M.S.; project administration, P.S. and M.S.

**Funding:** This research was financially supported by the Tyrolean Tourism Research Fund.

**Acknowledgments:** We would also like to express our gratitude to the three Tyrolean destination management organizations Hohe Salve, Innsbruck as well as Pitztal.

**Conflicts of Interest:** The authors declare no conflict of interest.

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
