# Peer review of "Does Health-Oriented Tourism Contribute to Sustainable Mobility?"

_sustainability, doi:10.3390/su11092633_

Round 1
Reviewer 1 Report
Dear Authors,
Thanks for letting me read this interesting manuscript. Sustainable mobility and health-related issues in an Alpine destination are highly relevant either to respect environmental sustainability and to enhance social sustainability in the region.
The manuscript is well structured and well written, for its clarity it is a joyful reading. The research questions are explained and analysed. Research methods are sound. Discussion, limitations and conlusions are clearly presented and well argumented.
Maybe the literature review could be enriched, I suggest checking Anna Scuttari's works published by the Journal of Sustainable Tourism.
Author Response
Dear Authors,
Thanks for letting me read this interesting manuscript. Sustainable mobility and health-related issues in an Alpine destination are highly relevant either to respect environmental sustainability and to enhance social sustainability in the region.
First of all, we would like to thank you for the valuable feedback and suggestions, which we have analysed carefully. We attempted to integrate your comments, considering also the suggestions of the other reviewers. The changes were made in the track changes mode so that you can easily follow the changes we have made in order to improve the manuscript.
The manuscript is well structured and well written, for its clarity it is a joyful reading. The research questions are explained and analysed. Research methods are sound. Discussion, limitations and conlusions are clearly presented and well argumented.
Thank you!
Maybe the literature review could be enriched, I suggest checking Anna Scuttari's works published by the Journal of Sustainable Tourism.
Thank you for pointing out this reference. We have now extended the literature review by integrating work from Anna Scuttari, that we believe, adds valuable content to our work. Please see for example page, X, line y

Reviewer 2 Report
Thank you for the opportunity of reviewing this interesting article. It addresses a topic of high importance for investigating the sustainability nexus in tourism.
I think the paper can be consistently improved by addressing the following points:
The abstract should be more focused and clearly reflect the objectives of the studies and main findings
The Introductiosn also should be oriented to display the main focus of the paper, the organization of the studies and the investigated issues
The literature part is quite poor and I suggest to remove general issues (such as definition of sustainability) and reflect the exact treatement of the topic in the existing literature and the identified gap that should be addressed
The writing can be improved
The final part should present the conclusions and also the utility of the findings. There is no reference to managerial / policy implications and this lack shuld be addressed.
Good luck!
Author Response
REVIEWER 2
Open Review
(x) I would not like to sign my review report
( ) I would like to sign my review report
English language and style
( ) Extensive editing of English language and style required
(x) Moderate English changes required
( ) English language and style are fine/minor spell check required
( ) I don't feel qualified to judge about the English language and style
Yes | Can be improved | Must be improved | Not applicable | |
Does the introduction provide sufficient background and include all relevant references? | ( ) | (x) | ( ) | ( ) |
Is the research design appropriate? | ( ) | (x) | ( ) | ( ) |
Are the methods adequately described? | ( ) | (x) | ( ) | ( ) |
Are the results clearly presented? | ( ) | ( ) | (x) | ( ) |
Are the conclusions supported by the results? | ( ) | ( ) | (x) | ( ) |
Comments and Suggestions for Authors
Thank you for the opportunity of reviewing this interesting article. It addresses a topic of high importance for investigating the sustainability nexus in tourism.
First of all, we would like to thank you for the valuable feedback and suggestions, which we have analysed carefully. We attempted to integrate your comments, considering also the suggestions of the other reviewers. The changes were made in the track changes mode so that you can easily follow the changes we have made in order to improve the manuscript.
I think the paper can be consistently improved by addressing the following points:
The abstract should be more focused and clearly reflect the objectives of the studies and main findings
Thank you for your comment. We agree and we have now shortened the abstract in general to be more concise in its rationale.
The Introduction also should be oriented to display the main focus of the paper, the organization of the studies and the investigated issues.
The literature part is quite poor and I suggest to remove general issues (such as definition of sustainability) and reflect the exact treatment of the topic in the existing literature and the identified gap that should be addressed.
Thank you for your comment. We removed the section on sustainability in tourism, which contained more general issues that gave our topic a wider context.We furthermore attempted to describe the research gap in a more distinct way, also by adding relevant literature (Scuttari). The intensified analysis of the literature follows in the second chapter of the paper (Theoretical Background).
The writing can be improved.
The article has been proof-read by a native speaker in order to improve the writing.
The final part should present the conclusions and also the utility of the findings. There is no reference to managerial / policy implications and this lack should be addressed.
Thank you for your comment. The implications of the study in hand can now be found in section 7 Implications and Outlook (lines 492 to 530)
Reviewer 3 Report
Overall, the article is well-written and presents an interesting case study. The chosen methods are appropriate and correspond to the research questions. However, I would recommend the following:
1. Shorten the abstract - it needs to be more concise and reveal more about the results of the study instead of giving details about the chosen methodological approach.
2. Justification of the case study: this is a major weakness. Why did you choose Tyrol? What makes it a good study?
3. Aim and objectives: you say mixed methods but all your questions suggest a qualitative approach? There are no hypotheses as such.
4. Sampling: How dod you choose the respondents for the focus groups? You have provided limited information that they have been recommend by the hotels. Why these hotels?
5. Survey vs focus groups: why did you choose both? What was the contribution and how was it different?
6. Limitations: Any?
Please also consider the following:
Line 11: Perhaps adding ‘previous’ at the beginning of the sentence would be helpful
Line 15: I am not sure if you can ‘close’ a gap with one article. Maybe you can ‘contribute’ or ‘address this gap’.
Line 17: Too much details for an abstract. Keep these for the methodology section.
Line 21: What do you mean by ‘front desk’? A hotel, a tourist information center?
Line 43: Cities such as…
Line 51: Some authors, such as?
Line 55: ‘Inadequate’ is quite a strong work here, consider replacing it.
Line 56-59: What were the findings of this project? Was it a research or?
Line 197-198: The ‘hypotheses’ are research questions and are very qualitative. I cannot personally see the quantitative element here, you need to formulate this in a better way.
Author Response
Comments and Suggestions for Authors
Overall, the article is well-written and presents an interesting case study. The chosen methods are appropriate and correspond to the research questions. However, I would recommend the following:
First of all, we would like to thank you for the valuable feedback and suggestions, which we have analysed carefully. We attempted to integrate your comments, considering also the suggestions of the other reviewers. The changes were made in the track changes mode so that you can easily follow the changes we have made in order to improve the manuscript.
1. Shorten the abstract - it needs to be more concise and reveal more about the results of the study instead of giving details about the chosen methodological approach.
Thank you for your comment. Regarding your advice we have now shortened the abstract in general to be more concise and consistent.
2. Justification of the case study: this is a major weakness. Why did you choose Tyrol? What makes it a good study?
We have added a more precise justification for the study region. As physically active people were of interest and the DMO want to promote physically active vacations, we believe that Tyrol represents the optimal region for this study. Please refer to the Methods section 4.1 for the detailed changes (lines 199 ff).
3. Aim and objectives: you say mixed methods but all your questions suggest a qualitative approach? There are no hypotheses as such.
The mixed methods approach is represented firstly by using focus groups (qualitative design) to receive a more in-depth picture of the topic under research and secondly by using a questionnaire (quantitative design) whose design was informed by the results of the focus groups. Thus, part of our methods used a qualitative design whilst others used a quantitative design. Independently of this design, you are right to state that we only have limited pre-defined hypotheses that we aim to test statistically. This is due to the innovative and explorative character of this study. The aim of the study was not to test and generalize pre-defined hypotheses as we cannot derive these on the basis of evidence, as there is limited evidence relating to this topic. The aim was rather to receive a first impression of the interdependencies between mobility patterns and physical activity of tourists in a sports-related region such as Tyrol, to compare these patterns with those at home and to descriptively evaluate whether the patterns at home change due to the vacation.
4. Sampling: How did you choose the respondents for the focus groups? You have provided limited information that they have been recommend by the hotels. Why these hotels?
Thank you for your comment. As described in line 226ff the participants in the focus groups were selected by the hotels, which, themselves, were suggested by the specific destination management organization and approached by us. However, we made participation dependent on having had a previous sojourn in the specified holiday region. So, the chosen sample represents a convenience sample. We added this information accordingly.
5. Survey vs focus groups: why did you choose both? What was the contribution and how was it different?
Thank you for your comment. As outlined above, there was no available validated questionnaire to answer our research questions. Thus, to ensure content validity, we held focus groups and used the results of these focus groups to inform the design of the questionnaire. We described this in more detail in section 4.2.
6. Limitations: Any?
Thank you for pointing this out. The limitations of the study at hand are described in section 6.3 Limitations.
Please also consider the following:
Line 11: Perhaps adding ‘previous’ at the beginning of the sentence would be helpful.
We have now added the suggested word.
Line 15: I am not sure if you can ‘close’ a gap with one article. Maybe you can ‘contribute’ or ‘address this gap’.
Thank you, we have now replaced ‘close’ by ‘address’.
Line 17: Too much details for an abstract. Keep these for the methodology section.
The abstract has now been remodelled and shortened.
Line 21: What do you mean by ‘front desk’? A hotel, a tourist information center?
We have now added ‘hotels’ for better understanding.
Line 43: Cities such as…
We have added information about the PASTA Project, which includes several examples.
Line 51: Some authors, such as?
The authors are mentioned in the citation brackets at the end of the sentence.
Line 55: ‘Inadequate’ is quite a strong work here, consider replacing it.
Thank you! This was the original wording in the cited reference, but, we have now replaced the word with ‘unsuitable’.
Line 56-59: What were the findings of this project? Was it a research or?
We have now added more information, therefore please see lines 64 to 76.
Line 197-198: The ‘hypotheses’ are research questions and are very qualitative. I cannot personally see the quantitative element here, you need to formulate this in a better way.
Thank you, we restructured the sentence to improve understanding. For a more precise explanation please see answer to issue 3.

Round 2
Reviewer 2 Report
Thank you for providing a revised version of your paper. My comments and suggestions were addressed and I can now endorse publication.